

# Effect of 8-hydroxyquinoline and derivatives on human neuroblastoma SH-SY5Y cells under high glucose

Wilasinee Suwanjang[1], Supaluk Prachayasittikul[2] and Virapong Prachayasittikul[3]

[1] Center for Research and Innovation, Faculty of Medical Technology, Mahidol University, Bangkok, Thailand
[2] Center of Data Mining and Biomedical Informatics, Faculty of Medical Technology, Mahidol University, Bangkok, Thailand
[3] Department of Clinical Microbiology and Applied Technology, Faculty of Medical Technology, Mahidol University, Bangkok, Thailand

## ABSTRACT

8-Hydroxyquinoline and derivatives exhibit multifunctional properties, including antioxidant, antineurodegenerative, anticancer, anti-inflammatory and antidiabetic activities. In biological systems, elevation of intracellular calcium can cause calpain activation, leading to cell death. Here, the effect of 8-hydroxyquinoline and derivatives (5-chloro-7-iodo-8-hydroxyquinoline or clioquinol and 8-hydroxy-5-nitroquinoline or nitroxoline) on calpain-dependent (calpain-calpastatin) pathways in human neuroblastoma (SH-SY5Y) cells was investigated. 8-Hydroxyquinoline and derivatives ameliorated high glucose toxicity in SH-SY5Y cells. The investigated compounds, particularly clioquinol, attenuated the increased expression of calpain, even under high-glucose conditions. 8-Hydroxyquinoline and derivatives thus adversely affected the promotion of neuronal cell death by high glucose via the calpain-calpastatin signaling pathways. These findings support the beneficial effects of 8-hydroxyquinolines for further therapeutic development.

## INTRODUCTION

Diabetes mellitus (DM) is a complex metabolic disorder featuring chronic hyperglycemia and tremendously impacts human health worldwide. Hyperglycemia contributes to the long-term diabetic complications i.e., retinopathy, nephropathy and neuropathy (*Aronsonons, 2008*). Epidemiological evidence suggests that patients with DM have a significantly high risk (50–100%) of developing Alzheimer's disease (*Biessels et al., 2006*). Diabetic patients exhibit cognitive impairment including damaged verbal memory, diminished mental speed and mental flexibility (*Cukierman, Gerstein & Williason, 2005*). Chronic hyperglycemia may accelerate the development of Alzheimer's disease, and many Alzheimer's patients exhibit impaired fasting glucose (*Janson et al., 2004*).

Neuronal cells cannot protect themselves from the harmful effects of excess glucose. The most likely mechanism for glucose toxicity is the generation of excess reactive oxygen species (ROS) via multiple mitochondrial and non-mitochondrial pathways (*Newsholme et al., 2007*). In addition to ROS production, high glucose levels trigger multiple biochemical

Corresponding author
Virapong Prachayasittikul,
virapong.pra@mahidol.ac.th

pathways and toxicity, which contribute to damage to DNA, lipid, proteins and subsequent neurotoxicity (*Li et al., 2014*). However, the mechanisms underlying the association of high glucose with neurodegeneration remain to be fully elucidated.

Calpain is an intracellular $Ca^{2+}$-dependent cysteine protease that is activated by increased intracellular $Ca^{2+}$. Calpain plays a vital role in glucose metabolism, cytoskeletal remodeling for cell cycle regulation and apoptosis, probably as a consequence of a loss of $Ca^{2+}$ homeostasis (*Vosler, Brennan & Chen, 2008*). Calpain cleaves and inactivates pro-caspase 9, pro-caspase 3 (*Chua, Guo & Li, 2000*) and APAF-1 (*Lankiewicz et al., 2000*). Calpastatin is a specific endogenous calpain inhibitor (*Croall & Ersfeld, 2007*). Calpain activity underlies the pathophysiology of several neurodegenerative diseases such as ischemia and epilepsy (*Vosler, Brennan & Chen, 2008*), and overexpression of calpastatin improves ischemia and reperfusion (*Maekawa et al., 2003*). Interestingly, the increase in calpain expression has been related to Bax, caspase-12, caspase-9 and caspase-3 in dopaminergic neurons (*Das, Banik & Ray, 2006*; *McGinnis et al., 1998*). The relative levels of calpain and dopamine in neuron involve the process of neurodegeneration such as Parkinson's and Alzheimer's diseases (*Chen, Nguyen & Sawmiller, 2011*; *Carragher, 2006*).

The biometal chelators (*Prachayasittikul et al., 2013*) 8-hydroxyquinoline, 5-chloro-7-iodo-8-hydroxyquinoline (clioquinol) and 5-nitro-8-hydroxyquinoline (nitroxoline), shown in (Fig. 1A), have been proposed as a potential therapeutic strategy for the treatment of Alzheimer's disease (*Bush, 2008*). Clioquinol was identified as a prototype metal-protein-attenuating compound (*Barnham, Cheny & Cappai, 2004*). The effect of clioquinol is related to its lipophilicity and ability to form relatively stable complexes with zinc (II) and copper (II) ions. Several reports have provided evidence that long-term pretreatment with clioquinol reduces the susceptibility of substantia nigra neurons to neurotoxin (*Kaur et al., 2003*). These compounds (Fig. 1A) are structurally related and bear 8-hydroxyquinoline as a core structure. Clioquinol is a halogenated derivative, and nitroxoline is the nitro derivative of 8-hydroxyquinoline. 8-Hydroxyquinoline and derivatives are bioavailable antioxidants that can cross the blood–brain barrier and inhibit metal-hydrogen peroxide production (*Barnham et al., 2004*).

Herein, the protective effects of 8-hydroxyquinoline, clioquinol and nitroxoline on human neuroblastoma cells under high glucose were investigated.

## MATERIALS AND METHODS

### Chemicals and reagents

Minimum essential medium (MEM), Ham's F-12 medium, fetal bovine serum (FBS), penicillin and streptomycin were purchased from Gibco BRL (Gaithersburg, MD, USA). Mouse monoclonal anti-actin (catalog number 3700), rabbit polyclonal anti-calpain (catalog number 2539), anti-calpastatin (catalog number 4146) and horseradish peroxidase-conjugated goat anti-mouse IgG and anti-rabbit IgG antibody were supplied by cell signaling (Beverly, MA, USA). Enhanced chemiluminescence (ECL) plus western blotting reagent was purchased from Amersham Biosciences (Piscataway, NJ, USA). The human dopaminergic neuroblastoma (SH-SY5Y) cell line was obtained from American Type Culture Collection

**A**

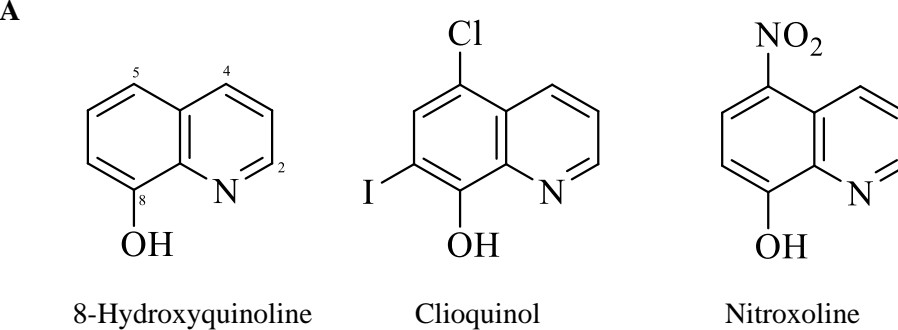

**B**

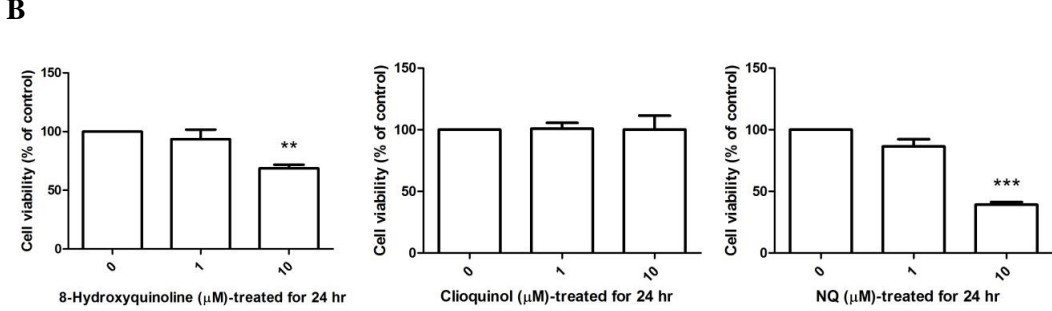

**Figure 1 8-Hydroxyquinoline, clioquinol and nitroxoline.** (A) Chemical structures. (B) Effect of 8-hydroxyquinoline and derivatives on cell viability in SH-SY5Y cells. Cells were treated with 8-hydroxyquinoline and derivatives at 1 and 10 $\mu$M for 24 h. Cell viability was measured using the MTT assay and is presented as the percentage of control cells. The results are expressed as the mean $\pm$ S.E.M. of four independent experiments. One-way analysis of variance (ANOVA) and the Tukey-Kramer multiple comparisons test were performed for statistical analysis. $**P < 0.01$ and $***P < 0.001$ compared with the control.

(Manassas, VA, USA). SH-SY5Y cells are a thrice cloned subline of bone marrow biopsy-derived line SK-N-SH. SH-SY5Y cell has dopamine-$\beta$-hydroxylase activity and express tyrosine hydroxylase. 8-Hydroxyquinoline (99%), clioquinol ($\geq$95%), and nitroxoline (96%) were purchased from Sigma-Aldrich (St Louis, MO, USA).

## Cell cultivation

SH-SY5Y cells (passage number less than 25) were cultured in 75-cm$^2$ flasks in MEM-F12 supplemented with 10% heat-inactivated FBS and 100 U/mL penicillin/streptomycin. Cells were maintained at 37 °C in an atmosphere of 5% CO$_2$ and 95% humidified air incubator, and were feed with medium every other day. To perform experiments, cells were seeded in 96-well and 6-well plates and grown to 70–80% confluence. Before the start of treatment, the medium was replaced with MEM-F12 containing 1% (v/v) FBS, as previously described (*Dayem et al., 2014*; *Kovalevich & Langford, 2013*). It has been shown that cell incubation with D-glucose for 24 h significantly induced cell apoptosis via the activation of c-Jun N-terminal protein kinase (JNK) and p-38 mitogen-activated protein kinase (MAPK) (*Chen*

*et al., 2013*; *Ho et al., 2000*). In case of glucose-treated cells for 2 h, the cells had significantly higher level of ROS accumulation and promoted apoptotic cell death (*Wu et al., 2004*). Up to date, the mechanism of high glucose contributing to degeneration in neuronal cells remains poorly understood. To investigate the mechanism of high glucose level involved in neuronal cells death, in this study, the cells were treated with D-glucose or D-mannitol at various concentrations (5.5, 30, 60 and 120 mM) for 2 or 24 h, and compared the percentage of cell viability. In some experiments, 8-hydroxyquinoline and derivatives were added to the medium for 2 h prior to an incubation with D-glucose for 24 h. Control untreated cells were incubated with the culture medium. Mannitol was utilized as an osmotic control.

## Cell viability assay

The 3-(4,5-dimethylthiazol-2-yl)-2,5-diphenyltetrazolium bromide (MTT) assay was used to assess neuronal injury after treatment of SH-SY5Y cells with a drug. When MTT is taken up by live cells, it is converted from yellow to dark blue formazan crystals by cellular dehydrogenase (*Stockert et al., 2012*). MTT in 0.1 mM phosphate buffered saline (PBS) was added to each well and incubated at 37 °C for 4 h. The solution was discarded, and extraction buffer (0.04 N HCl in isopropanol) was added. The optimal densities were measured at a spectral wavelength of 570 nm using a microtiter plate reader.

## Western immunoblotting

Treated cells were harvested and lysed by adding lysis buffer and scraped off the plate. Cells were sonicated for 10 s and centrifuged for 15 min at 12,000 g. The supernatants were collected and separated by sodium dodecyl sulfate-polyacrylamide gel electrophoresis. The protein bands were transferred to nitrocellulose membranes and washed with Tris-buffered saline and Tween20 (TBST) for 5 min. The membranes were incubated in a blocking buffer (5% non-fat dry milk in TBST), then washed with TBST and incubated in primary antibodies at 4 °C overnight. After the incubation, the membranes were washed three times with TBST for 5 min and then incubated in HRP-conjugated secondary antibody for 1.5 h, followed by washing three times for 5 min each time with TBST. The blots were developed with ECL Plus Western Blotting detection reagents.

## Immunocytochemical analysis

SH-SY5Y cells were seeded on sterile glass coverslips at 37 °C for 24 h and then exposed to D-glucose in the medium containing 1%FBS for 24 h; control cells were incubated with medium for 24 h. The cells were incubated with MitoTracker® Red CMXRos for 30 min. The medium was removed, and the cells were washed with ice-cold PBS. The cells were fixed with 4% paraformaldehyde in PBS for 30 min at 4 °C and washed with PBS three times for 5 min each time. Cells were permeabilized with 1% Triton X-100 in PBS for 10 min at room temperature and rinsed with PBS three times. Non-specific antibody binding sites were blocked by incubating the cells with 10% donkey serum in PBS containing 0.3% Triton X-100 and 1% bovine serum albumin (BSA) for 10 min at room temperature. Cells were incubated with the primary antibody against calpain (1:1,000 in PBS containing 0.3% Triton X-100 and 0.25% BSA) overnight at 4 °C, followed by incubation with fluorescein isothiocyanate (FITC)-conjugated donkey anti-rabbit IgG (1:200 in PBS containing 0.3%

Triton X-100 and 0.25% BSA) for 2 h at room temperature. The cells were washed three times with PBS, and stained slides were mounted using antifade reagent in glycerol buffer (Vector Laboratories, Burlingame, USA) and visualized by fluorescence microscopy (Olympus, Tokyo, Japan).

### Statistical analysis

Data are expressed as mean $\pm$ S.E.M. Significance was assessed by one-way analysis of variance (ANOVA) followed by a Tukey-Kramer test using SPSS 18 software package for Windows (Chicago, IL, USA). Probability ($P$) values of less than 0.05 were considered statistically significant.

## RESULTS

### Effect of high glucose on cell viability of SH-SY5Y cells

The effect of high-glucose exposure on cell viability was investigated in SH-SY5Y cells using various concentrations of D-glucose and D-mannitol (an osmolality control) medium for 2 h and 24 h. Treatment with D-glucose for 2 h significantly decreased cell viability to 91.31 $\pm$ 0.73% at 60 mM ($P < 0.01$) and to 82.59 $\pm$ 2.59% at 120 mM ($P < 0.001$) compared with normal medium (5.5 mM glucose) ($F$-value = 30.779) whereas treatment with D-glucose for 24 h significantly decreased cell viability to 89.10 $\pm$ 3.23% at 30 mM ($P < 0.05$), to 78.48 $\pm$ 1.16% at 60 mM ($P < 0.001$), and to 73.97 $\pm$ 2.31% at 120 mM ($P < 0.001$) ($F$-value = 31.564) (Fig. 2A). To rule out an effect of osmotic stress on SH-SY5Y cells treated with high glucose, cells were incubated with D-mannitol under the same conditions for the indicated time. The differences in cell viability, between cells treated with D-glucose and with D-mannitol at 60 or 120 mM for 24 h were statistically significant. However, high glucose at 60 and 120 mM induced neuronal cell death as a result of hyperglycemia and hyperosmolarity. A decrease in the cell viability of neuronal cells was noted when the cells were treated with high glucose for 2 h. Increasing the ambient D-glucose concentration caused dose- and time-dependent decreases in cell viability. Thus, 120 mM D-glucose was selected to treat neurons in this study because this concentration has been used in many studies of hyperglycemia *in vitro* (*Haslinger et al., 2001*; *Li, Zhang & Sima, 2003*; *Song et al., 2015*).

### Effect of high glucose induced calpain and reduced calpastatin protein levels

To determine if the increase in calcium-dependent pathways induced by high glucose treatment occurs via upregulation of calpain protein, SH-SY5Y cells were incubated with various glucose concentrations (5.5–120 mM) for the indicated time, the cell lysate was collected, and calpain and calpastatin levels were determined by Western blot analysis. Treatment with 120 mM D-glucose for 2 h or 24 h significantly increased calpain levels by 129.69 $\pm$ 8.30% ($P < 0.01$) ($F$-value = 7.031) and 134.44 $\pm$ 3.97% ($P < 0.001$) ($F$-value = 31.964) compared with control cells at the same time points (Fig. 2B). These results demonstrate that high glucose induced calpain expression.

Further investigation of calpastatin, a specific endogenous calpain inhibitor, was performed by Western immunoblotting. Interestingly, exposure to 60 mM D-glucose

**A**

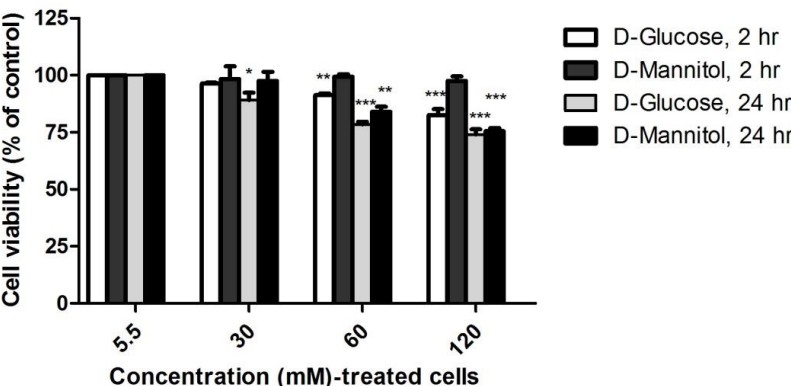

**B**

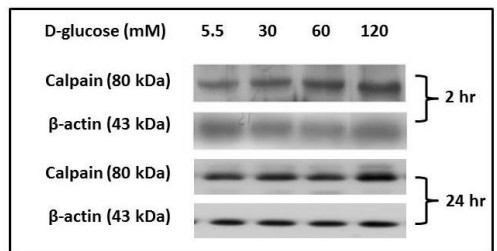
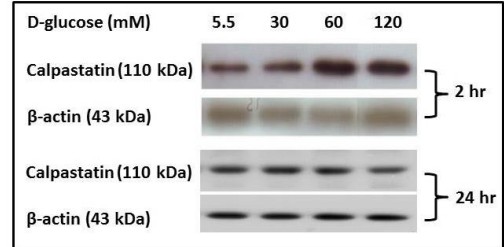

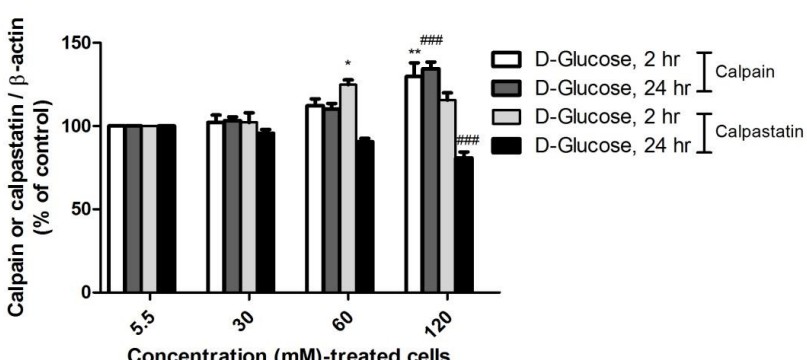

**Figure 2 High glucose-induced alteration of cell viability, capain and capastatin proteins expression.** Cells treated with D-glucose concentrations (30, 60 and 120 mM) for 2 h and 24 h were compared to cells treated with control medium containing 5.5 mM D-glucose and mannitol as an osmotic control. (A) Cell viability was measured using the MTT assay. (B) The levels of calpain and calpastatin were determined by Western blot analysis. Protein bands were quantified by densitometry, and their differences are represented in the graph as the ratio of calpain and calpastatin to $\beta$-actin. The results are expressed as the mean + S.E.M. of four independent experiments. One-way analysis of variance (ANOVA) and Tukey-Kramer multiple comparisons test were performed for statistical analysis, $*P < 0.05$, $**P < 0.01$ and $***P < 0.001$ compared with the control at 2 h and $^{###}P < 0.001$ compared with the control at 24 h.

resulted in increased calpastatin expression as early as 2 h (124.80 ± 2.88%, $P < 0.01$) ($F$-value = 9.010), whereas 120 mM D-glucose exposure for 24 h significantly decreased calpastatin protein levels to 75.07 ± 4.35% ($P < 0.001$) ($F$-value = 11.909) compared with the control (Fig. 2B).

To demonstrate that the observed increase in calpain was related to cell death, an immunofluorescent double-labeling experiment was performed using MitoTracker Red as the mitochondrial marker. Control (5.5 mM D-glucose) cells exhibited weak immunostaining of calpain. However, cells treated with 120 mM D-glucose displayed a bright green speckled appearance that became more intense by 24 h after glucose administration (Fig. 3). Thus, exposure to high glucose resulted in an induction of calpain immunofluorescence staining in SH-SY5Y cells.

## Cytotoxicity of 8-hydroxyquinoline and derivatives

The cytotoxic effects of 8-hydroxyquinoline and derivatives on cultured cells were assessed at different concentrations using the tetrazolium salt reduction (MTT) assay. No significant cytotoxic effect of 8-hydroxyquinoline and derivatives were evident at 1 μM in SH-SY5Y cells (8-hydroxyquinoline: 93.52 ± 8.15% ($F$-value = 10.726); clioquinol: 100.8 ± 4.73% ($F$-value = 0.40); nitroxoline: 86.44 ± 5.87% ($F$-value = 78.113)) as shown in (Fig. 1B). Cytotoxic effects of 10 μM 8-hydroxyquinoline (68.67 ± 6.37%) and nitroxoline (39.17 ± 2.18%) were observed after treatment for 24 h, and therefore 1 μM was used in subsequent experiments.

## Protective effect of 8-hydroxyquinoline and derivatives on high glucose-reduced cell viability

The effects of 8-hydroxyquinoline and derivatives were further investigated by monitoring cell viability changes in response to high-glucose (120 mM) treatment for 24 h. Exposure to 1 μM clioquinol (93.35 ± 0.89%, $P < 0.001$) or nitroxoline (95.72 ± 0.92%, $P < 0.001$) significantly increased cell viability compared with high glucose-treated cells (73.97 ± 2.31% $P < 0.01$) ($F$-value = 24.262) (Fig. 4A). However, pretreatment with 1 μM 8-hydroxyquinoline also significantly increased cell viability to 86.89 ± 3.06%. The protective effect of the compounds in order of potency was nitroxoline > clioquinol > 8-hydroxyquinoline.

## Effect of 8-hydroxyquinoline and derivatives on high glucose-induced calpain-calpastatin alteration

8-Hydroxyquinoline and derivatives have been reported to exert antidiabetic activity (*Prachayasittikul et al., 2013*). Calpains are important regulators of the cell cycle and apoptosis, and their activities are dependent on the concentration of calcium in cells. We previously demonstrated that dexamethasone induced neuronal cell death via a calpain-dependent pathway (*Suwanjang et al., 2013*). In the present study, the effect of 8-hydroxyquinoline and derivatives on high glucose-induced calpain activation was observed (Fig. 4B). Treatment of SH-SY5Y cells with 120 mM D-glucose for 24 h resulted in calpain expression. Pretreatment with 1 μM 8-hydroxyquinoline and derivatives significantly attenuated calpain expression (8-hydroxyquinoline; 109.82 ± 5.28% ($P < 0.05$) ($F$-value = 11.489); clioquinol; 104.91 ± 4.95% ($P < 0.01$) ($F$-value = 13.919); nitroxoline: 105.47 ±

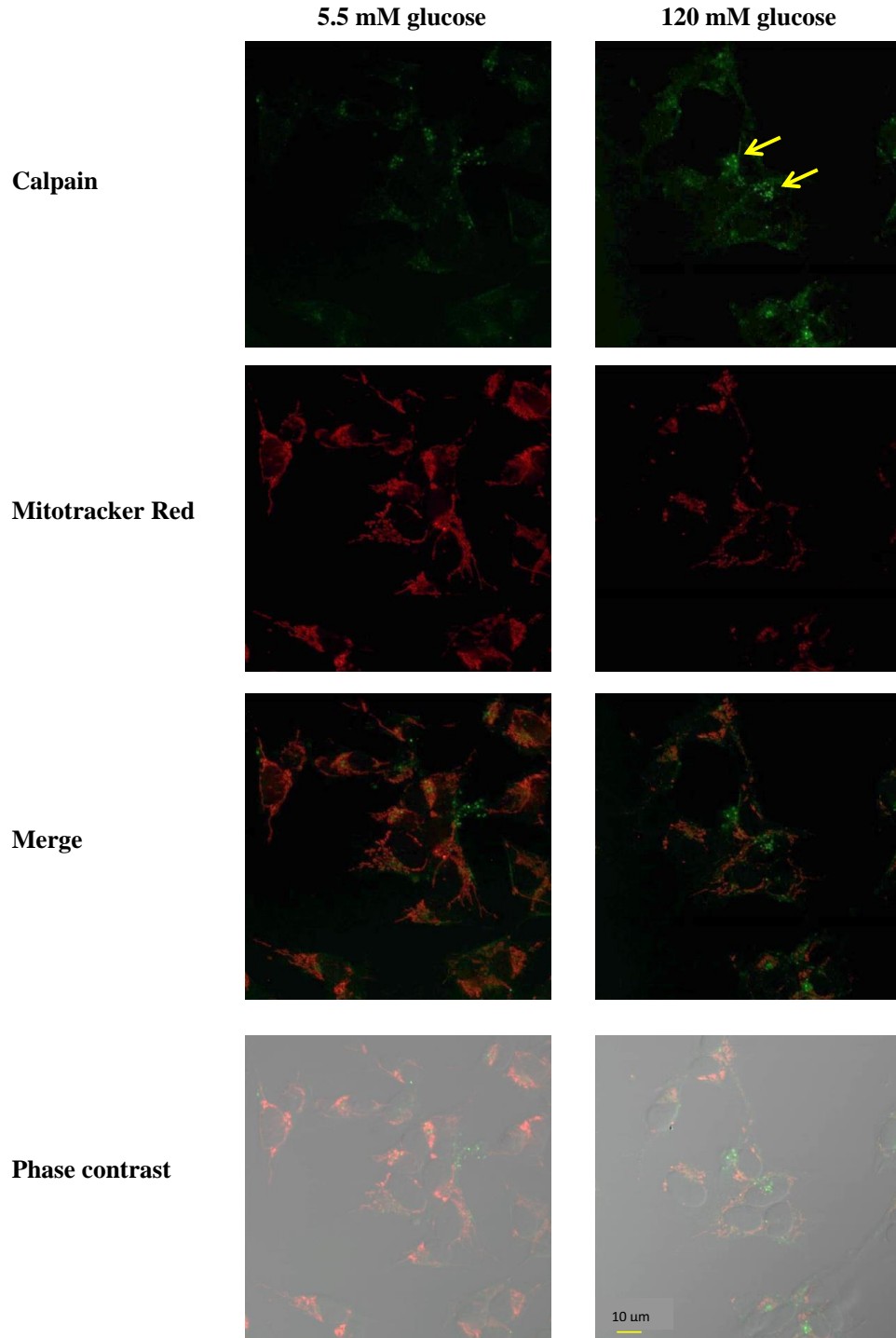

**Figure 3 Imaging microscopic analysis of SH-SY5Y cells demonstrating the D-glucose-induced increase in calpain expression.** Cells were treated with 120 mM D-glucose for 24 h. The control cells were incubated with the culture medium for 24 h. The green color indicates calpain immunostaining using fluorescein-5-isothiocyanate (FITC)-conjugated anti-IgG.
**A**

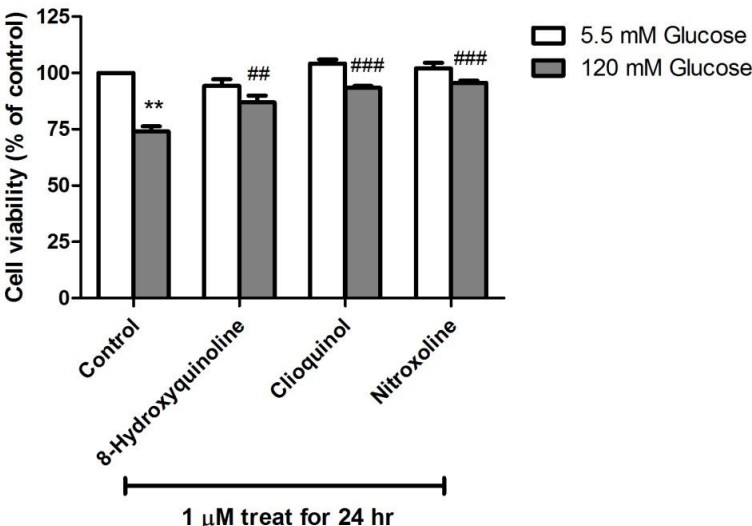

**B**                                                    **C**

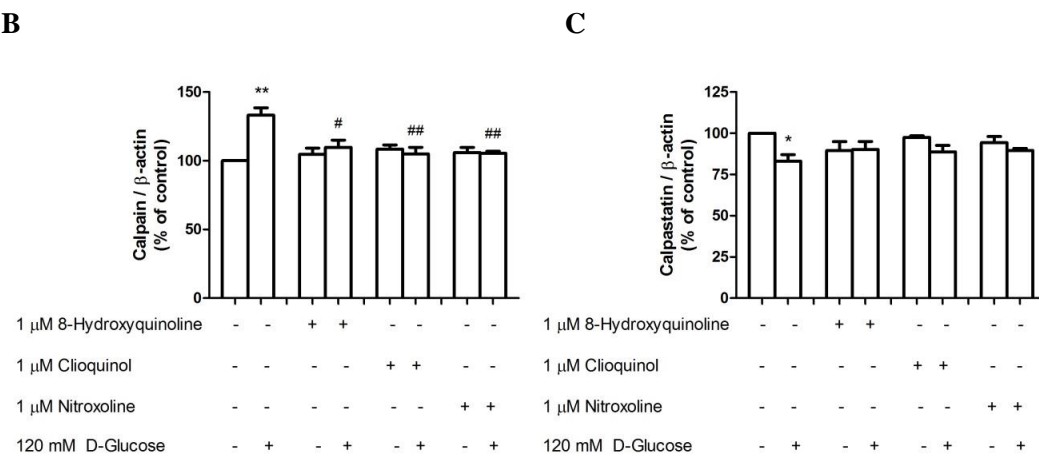

**Figure 4** **The effect of 8-hydroxyquinoline and derivatives on the high glucose (120 mM) in SH-SY5Y cells.** Cells were treated with high glucose for 24 h. Some cells were pre-treated with 1 µM 8-hydroxyquinoline and derivatives for 2 h prior to incubation with 120 mM high glucose for another 24 h. The control cells were incubated with the culture medium for 24 h. (A) Cell viability was measured using the MTT assay. The results are expressed as the mean ± S.E.M. of four independent experiments. (B) Calpain and (C) calpastatin expressions were determined by Western blot analysis. Protein bands were quantified by densitometry, and the changes are represented in the graph. Calpain and calpastatin expressions are presented as the ratios of calpain or calpastatin/$\beta$-actin protein bands. The results are expressed as the mean ± S.E.M. of three independent experiments. One-way analysis of variance (ANOVA) and the Tukey-Kramer multiple comparisons test were performed for statistical analysis. *$P < 0.05$, **$P < 0.01$ and ***$P < 0.001$ compared with the control and #$P < 0.05$, ##$P < 0.01$, ###$P < 0.001$ compared with high glucose-treated cells.

1.49% ($P < 0.01$) compared with the high glucose-treated cells ($133.19 \pm 5.32\%$, $P < 0.001$) ($F$-value $= 19.840$). A greater protective effect was observed for clioquinol, as evidenced by lower calpain expression under high glucose treatment. However, the protective effect of nitroxoline was comparable to that of clioquinol. By contrast, 8-hydroxyquinoline and derivatives tended to increase the expression of the calpain inhibitor (calpastatin, Fig. 4C) by high glucose (8-hydroxyquinoline: $90.13 \pm 4.93\%$ ($F$-value $= 2.840$); clioquinol: $88.77 \pm 3.88\%$ ($F$-value $= 7.683$); nitroxoline: $89.61 \pm 1.31\%$ ($F$-value $= 6.570$)) compared with the high glucose-treated cells ($83.03 \pm 4.02\%$, $P < 0.05$). Moreover, treatment with 8-hydroxyquinoline and derivatives had no significant effects on the expressions of calpain and calpastatin in untreated control cells.

## DISCUSSION

Hyperglycemia is considered a risk factor of neurodegenerative diseases (*Kopf & Frolich, 2009*). Impairments in signaling mechanisms contribute to increased neuronal cell death. Numerous studies have focused on elucidating the mechanism by which high glucose toxicity enhances death mechanisms. The optimal concentration of glucose for neuronal survival is reportedly in the range of 25–30 mM. Here, cell viability under high-glucose exposure in human neuroblastoma SH-SY5Y cells was investigated. The mechanisms underlying hyperglycemia and hyperosmolarity have been studied extensively. During hyperglycemia, high levels of glucose-induced oxidative stress can cause cellular damage. In addition, excess glucose leads to neurotoxicity via increased apoptosis and inhibition of proliferation. This may activate p38 kinase associated with apoptosis via protein kinase C-dependent and -independent pathways (*Igarashi et al., 1999*). The results suggest that elevated glucose level initiates harmful mechanisms leading to neuronal cell degeneration (neuropathy). High glucose (120 mM) was reported to affect $Ca^{2+}$ homeostasis (*Kimura, Oike & Ito, 1998*). It is also well established that high glucose (120 mM) induced oxidative stress and promoted calcium influx in a variety of cell types including human monocytes (*Wuensch et al., 2010*) and cardiac cells (*Kumar, Kain & Sitasawad, 2012*; *Ozdemir et al., 2005*; *Cai et al., 2002*).

Impairment of $Ca^{2+}$ homeostasis is an important factor in the development of neuronal degeneration (*Todorovic & Jevtovic-Todorovic, 2014*). Under physiological conditions, calpain is localized in the cytosol and is in an inactive form in the absence of calcium. Calpain is activated by cytosolic $Ca^{2+}$ overload. The dysregulation of intracellular calcium levels is an indicator of neuronal injury through the activation of several enzymes such as calpains and phospholipases as well as mitochondrial alterations (*Araujo, Verdasca & Leal, 2004*). The calpain system plays a major role in various cellular signaling processes, including signal transduction, cell adhesion and motility, cell growth, differentiation and cell death. Calpain activates both caspase-dependent and caspase-independent pathways to promote apoptosis. In the apoptotic pathway, calpain cleaves apoptotic inducing factor, which activates DNA degradation (*Baritaud et al., 2010*). Thus, the activation of calpain may have an important role in many diseases such as retinal photoreceptor apoptosis (*Mahajan et al., 2012*) and ischemia (*Rami, 2003*). A high concentration of glucose also results in morphological alterations and cell death via processes related to the apoptotic

pathway (*Allen, Yaqoob & Harwood, 2005*). Accumulation of oxidative stress is present in diabetes. Therefore, high glucose can induce cellular hypertrophy by excessive production of ROS. Clinical studies also indicate that high glucose enhances the pathology of diabetes by increasing oxidative stress (*Trombetta et al., 2005*). In general, ROS are recognized as a main source of molecular damage in hyperglycemia (*Piconi et al., 2006*).

Calpain activity is inhibited by endogenous calcium-dependent interactions with calpastatin. Calpastatin binds to the active site and inhibits calpain in the presence of calcium (*Goll et al., 2003*). Specific calpain inhibitors reduce neuronal damage in a number of different systems. The current findings demonstrate that high concentrations of glucose can lead to increased calcium levels and enhance calpain protein expression in a dose- and time-dependent manner. Increased calpain expression has been implicated in vascular inflammation and endothelial leakage in diabetes (*Scalia et al., 2007*). In addition, calpain plays significant roles in apoptotic processes (*Raynaud & Marcilhac, 2006*). However, the expression of proteins related to the specific calpain inhibitor (calpastatin) was decreased in SH-SY5Y cells treated with high glucose concentrations. Furthermore, previous studies have suggested that neuronal calpain activity mediates the initiation and expression of methamphetamine- and dexamethasone-induced cell death (*Suwanjang et al., 2013*; *Suwanjang et al., 2012*). Several mechanisms have been proposed for the effects of calpain during cell death, including cleavage of pro-caspase 3 and degradation of apoptotic proteins (*Camins et al., 2006*).

Several studies have revealed a functional association of 8-hydroxyquinoline and derivatives with cancer, inflammation, Alzheimer's and Parkinson's diseases (*Mao & Schimmer, 2008*). Of the tested derivatives, clioquinol is a bioavailable ligand with moderate affinity for copper, zinc and iron. Clioquinol possesses relatively high lipophilicity and crosses the blood–brain barrier. Clioquinol has been observed in brain tissue and cerebrospinal fluid (*Bondiolotti et al., 2007*) and exhibits neuroprotective effects in MPTP mouse (*Kaur et al., 2003*) and Alzheimer's models (*Suh, Jensen & Jensen, 2000*). Nitroxoline, a nitro derivative of hydroxyquinoline, is used as an antibacterial drug in patients with urinary tract infections (*Wagenlehner et al., 2014*; *Ghoneim, El-Desoky & Abdel-Galeil, 2011*). Recently, clioquinol and nitroxoline have been reported to exert anticancer activity against cholangiocarcinoma cells (*Chan-On et al., 2015*). Clinical trial data suggest that 8-hydroxyquinoline and its derivatives may also have benefits in preventing the development and progression of neurodegeneration. Clioquinol protects against cell death in *in vivo* and *in vitro* models of Parkinson's disease (*Wilkins et al., 2009*). The induction of calpain in neuronal cells might be closely related to several toxicity mechanisms, including caspase-3 activation (*Bastianetto et al., 2011*) and oxidative stress (*Suwanjang et al., 2013*). 8-Hydroxyquinoline and derivatives have been reported as potent antidiabetic agents (*Prachayasittikul et al., 2013*). The present study demonstrates that treatment of SH-SY5Y cells with 8-hydroxyquinoline and derivatives results in decreased calpain expression and reduced neuronal cell death after high glucose toxicity. Conversely, high-glucose toxicity might be controlled by treatment with 8-hydroxyquinoline and derivatives. Notably, the compounds reduced the expression of calpain in the order clioquinol ≈ nitroxoline > 8-hydroxyquinoline. It has been suggested that 8-hydroxyquinoline and derivatives, especially clioquinol prevent $Ca^{2+}$ influx and

calcium signal in Alzheimer's (*LeVine et al., 2009*; *Abramov, Canevari & Dunchen, 2005*; *Lin, Bhatia & Lal, 2003*; *Cherny et al., 2001*) and Parkinson's diseases (*Cacciatore et al., 2013*). Furthermore, it is also reasonable that 8-hydroxyquinoline and derivatives decrease calpain but increase calpastatin expressions via their antioxidant activities and reduce intracellular calcium level in high glucose toxicity. In addition, the decrease in calcium level may be enhanced by metal chelating property of 8-hydroxyquinoline and derivatives (*Prachayasittikul et al., 2013*).

## CONCLUSIONS

This study reveals that 8-hydroxyquinoline and derivatives offer partial neuroprotection against high glucose toxicity and modulate the balance between calpain and calpastatin expressions. These findings provide a foundation for the further therapeutic development of 8-hydroxyquinoline compounds.

## ACKNOWLEDGEMENTS

We thank Ms.Chayanit Sirisuwat for collecting the samples.

### Funding

This project was supported by Mahidol University, Thailand, to Wilasinee Suwanjang, and by the Office of the Higher Education Commission, Mahidol University, under the National Research University Initiative. The funders had no role in study design, data collection and analysis, decision to publish, or preparation of the manuscript.

### Grant Disclosures

The following grant information was disclosed by the authors:
National Research University Initiative.

### Competing Interests

The authors declare there are no competing interests.

### Author Contributions

- Wilasinee Suwanjang conceived and designed the experiments, performed the experiments, analyzed the data, contributed reagents/materials/analysis tools, wrote the paper, prepared figures and/or tables.
- Supaluk Prachayasittikul and Virapong Prachayasittikul conceived and designed the experiments, reviewed drafts of the paper.

### Data Availability

The raw data has been supplied as Supplementary Files.

## Supplemental Information

Supplemental information for this article can be found online at http://dx.doi.org/10.7717/peerj.2389#supplemental-information.

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
