# Peer review of "Effect of 8-hydroxyquinoline and derivatives on human neuroblastoma SH-SY5Y cells under high glucose"

_PeerJ, doi:10.7717/peerj.2389_

## Round 0.1 · original submission · Minor Revisions

Dear Authors,

Please heed the suggestions given by the two peer reviewers to improve your manuscript. Also note the annotated manuscript from Reviewer 1.

Please send the revised manuscript for English editing using a journal editing service (e.g. American Journal Editors or equivalent) - I would ask that you submit a certificate indicating that the English grammar and syntax has been revised.

Thanking you

Reviewer 1 ·

Basic reporting

Overall, the article was written in good standard English. Sufficient introduction is included in the article, citing relevant prior literature. Figures included are relevant to the content of the article and well-presented. Throughout the article,there are several typo errors encountered, particularly in the reference list.

Experimental design

The experimental design is suitable to answer the research question, which is to determine the neuroprotective effects of 8-hydroxyquinoline and its derivatives in high glucose condition.

In methods section, some parts could be improved by adding details e.g. concentrations of D-glucose and mannitol used in assays. Although these details are available on figures and figure legends, it would be more clear to readers if such information is included in the methods as well (parts that need to be clarified further are annotated in the pdf file). Also, it is not indicated in the article on the passage number of the SH-SY5Y used in the study.

Validity of the findings

Data presented in the article are clear and statistically sound, with the conclusions appropriately stated.

Annotated reviews are not available for download in order to protect the identity of reviewers who chose to remain anonymous.

Reviewer 2 ·

Basic reporting

No Comments

Experimental design

No Comments

Validity of the findings

No Comments

Additional comments

The manuscript entitle "Effect of 8-hydroxyquinoline and derivatives on neuroblastoma cells under high glucose" is well attempted to prove effective role of 8-hydroxyquinoline and derivatives. The manuscript was well written. However, author needs to do minor correction before considering for publication.
Title is not clear, so specific cell type should be mentioned.
In introduction, author should write few words interaction between dopaminergic neurons and calpain-calpastain. Besides, need to write Parkinson disease and calcium homeostasis.
In methods were written well however references should be added for cell cultivation procedure. The primary antibodies catalog number must be texted. Author did not say that why selected 2 and 24hr only. Also, how author select these particular D-glucose doses for this current study. D-glucose doses did not mentioned in methods at 2 or 24hr
In results, author must include F-values along p value.
Two references is too old (Jacobs 1978 and Mosmann1983), add recent references instead of these.
In figure 3, there is no scale bar; Legends for figure 2 and 4 are too lengthy, better to shorten. In the manuscript, legends are represented two times. It would be better to remove.
In discussion, how high glucose cause oxidative stress and increased calcium level must be given in details.
8-Hydroxyquinoline and derivatives reduced calpain in the present study. The details mechanisms did not write in discussion. Both 8-hydroxyquinoline and derivatives directly reduced calcium level or control oxidative stress or in another way increased anti-oxidant enzymes, this is should be highlighted with references.
Is there any previous study reported that 8-Hydroxyquinoline and derivatives reduced intracellular calcium level or the derivatives could block or inhibit calcium channels.
General question, normal physiological functions or even in diseased conditions, intracellular calcium regulated or dysregulated by many factors inside the cell. In this condition, what could be signaling to activate calpain or what could be other factors involved to suppress calpain activation in high calcium level in cells.

---

## Round 0.2 · accepted · Accept

Dear Authors, The manuscript is accepted - congratulations.

However, a few typos remain which you should fix in production

Reviewer 1 ·

Basic reporting

No further comments

Experimental design

No further comments

Validity of the findings

No further comments

Additional comments

The authors have made necessary changes to the manuscript. However, there are still couple of minor errors (typos) need to be made. Please refer to the annotated manuscript.

Annotated reviews are not available for download in order to protect the identity of reviewers who chose to remain anonymous.

Reviewer 2 ·

Basic reporting

No

Experimental design

No

Validity of the findings

No

Additional comments

Author did all necessary corrections and addressed all queries.